# Health and Wellness Projects Created by Student Pharmacists during Advanced Pharmacy Practice Experiences: Exploring the Impact on Professional Development

**DOI:** 10.3390/pharmacy12010005

**Published:** 2023-12-28

**Authors:** Laurie L. Briceland, Megan Veselov, Courtney Caimano Tackes, Jennifer Cerulli

**Affiliations:** Department of Pharmacy Practice, Albany College of Pharmacy and Health Sciences, Albany, NY 12208, USAcourtney.tackes@acphs.edu (C.C.T.); jennifer.cerulli@acphs.edu (J.C.)

**Keywords:** Community Advanced Pharmacy Practice Experiences, critical reflection, health campaigns, health and wellness promotion, pharmacy education, population health, professional identity formation, self-determination theory, student professional development

## Abstract

A curricular expectation of pharmacy educators is to equip students with strategies for the promotion of health and wellness. The impact on student professional development with involvement in such health promotions has been sparsely documented. The specific aim of this project was to explore the impact on student learning and professional development when they create, implement, and reflect upon a Health and Wellness Project (HWP). In 2022–2023, each student completing a Community Advanced Pharmacy Practice Experience created and implemented an HWP with the goal of serving as a “health promoter” (205 projects). A multi-method design of quantitative and qualitative assessment techniques was used to analyze the impact of creating/implementing the HWP on students’ learning, with a self-determination theory (SDT) framework utilized to evaluate professional development. Upon review, all projects met the acceptability criteria. Qualitative data analysis from a subset of 48 students identified themes of impact on learning, which included knowledge acquisition, enhancement of communication skills, opportunity for patient-centered interaction, selection of targeted educational strategies, and immersion into the role of health promoter. All three components of SDT were found to support professional development: competence in the field; relatedness to patients and the profession; and autonomy in creating the HWP. Student quotations demonstrated strong professional identity formation as students began to think, act, and feel like pharmacists.

## 1. Introduction

A longstanding curricular expectation of pharmacy educators is to equip students with strategies for the promotion of health and wellness within at-risk populations in the community [1,2,3,4]. Specifically, the most recent Doctor of Pharmacy educational outcomes (COEPA 2022) state: “Population Health and Wellness (Promoter): the learner will assess factors that influence the health and wellness of a population and develop strategies to address those factors”. Students are expected to utilize varied verbal and written communication strategies to educate diverse audiences while applying an innovative, creative mindset [3]. Over the past two decades, studies have demonstrated the value of student pharmacists in promoting public and preventive health in targeted populations/disease states (e.g., aspirin use in diabetic patients) or in specific initiatives (e.g., safe medication disposal) during Community Advanced Pharmacy Practice Experiences (APPEs) [5,6,7,8,9,10]. In this body of literature, we noticed a few gaps, specifically: (i) preceptors/faculty assigned previously developed projects to students instead of having students select a topic and create the project (supporting autonomy and the creative mindset); and (ii) the impacts of project participation on the students’ professionalization were sparsely, if at all, reported. In two recent studies, information on student professionalization was included. In a campus-based health campaign, student pharmacists conducting the effort were surveyed and reported that the activities provided a meaningful learning experience and a “high level of professional achievement and satisfaction” [11]. A study examining student perceptions of Community APPE participation reported that rotations that exposed students to “sufficient educational and wellness services” improved their professional development composite score [12].

Our Community APPE Health and Wellness Project (HWP) initiative builds on the published literature by enabling students to apply an innovative/creative mindset and document the professionalization impacts of participation in the HWP. Specifically, our 4th-professional year (P4) students are required to select a topic and create/implement an HWP as a component of the required Community APPE. Students provide a summary of their project and use guided self-reflection to describe the impact. Our study delves deeper to elucidate specific learning and professionalization impacts on students’ creation and delivery of a health and wellness initiative in a target population. Did our students experience autonomy and apply a creative mindset when implementing a Health and Wellness Project of their choice/interest? What competencies and relationships did the student develop? How did the student grow professionally? And what transformational behaviors do they hope to carry forward into their future career? The specific aim of this research project was to explore the impact on student learning and professional development when they create, implement, and reflect upon a Community APPE Health and Wellness Project.

## 2. Methods

This Community APPE HWP was conducted at Albany College of Pharmacy and Health Sciences, a private college in New York that offers a traditional 4-year Doctor of Pharmacy program. An accelerated 3-year Doctor of Pharmacy program was also in place during the study period and has since been phased out. The project underwent an Institutional Review Board review and met the criteria for exemption from the requirements of federal regulations. Over the past nine academic years, in accordance with the Center for the Advancement of Pharmacy Education (CAPE) 2013 Outcomes [2], each P4 student created and implemented a targeted HWP with the overarching goal of assuming the role of “health promoter” during their required Community APPE. Approximately one-third of students elected to complete a second Community APPE, in which an additional different HWP was completed. This research study will report on the most recently completed 2022–2023 cycle in which 157 students completed 205 HWPs over the course of eight 6-week APPE modules. 

### 2.1. Description of the Health and Wellness Project (HWP) Learning Activity 

The Accreditation Council for Pharmacy Education (ACPE) Standards 2016 [13] state the following requirements for Standard 2.3 (Health and Wellness) for the Doctor of Pharmacy program: ‘The graduate must be able to design prevention, intervention, and educational strategies for individuals and communities to manage chronic disease and improve health and wellness’. To address this standard, we designed a HWP learning activity to be completed by each student during their required 6-week Community APPE (and during a second Community APPE, if scheduled). For the HWP, each student was required to design and execute a targeted population-centric project promoting health and wellness in the community. In addition to meeting ACPE Standard 2.3 Health and Wellness, in 2021, we revised our HWP criteria such that each project was also required to address at least three additional CAPE [2] or ACPE educational standards [13] of the student’s choosing from options list in Table 1.

The intent for including these additional requirements was to enable students to strengthen their HWPs and to potentially address gaps noted in the literature. For instance, a comprehensive review assessing the published literature on student learning during APPEs noted that there were no publications identifying patient advocacy or innovative mindset as the primary outcome of focus [14]. Additionally, a 2021 survey [15] of US and Canadian Colleges of Pharmacy assessing the inclusion of cultural competency/social determinants of health and health literacy in pharmacy curriculums reported that fewer than 20% of survey respondents included these concepts in required APPEs, concluding that expansion of these topics within APPEs was warranted.

Preceptors for the students’ Community APPEs were provided guidance on the HWP and their role, including a Frequently Asked Questions (FAQ) document (see Document S1 FAQs for Preceptors in Appendix A). In consultation with preceptors, students were to select their topic for the HWP by the end of Week #1 (of their 6-week APPE) and submit it to Experiential Education (EE) for approval. Example photos from previous projects were available to students in EE Orientation and bulletin boards. Suggested topics for the HWP were provided to students and are listed in the Appendix B. After completion of the HWP within the Community APPE, students were to upload the following materials: (i) a complete description of the project, including objectives, references cited, location, personnel, time frame, and advertising; (ii) materials used in the project including audio and visual aids, printed copies of any handouts, pamphlets, and/or educational materials disseminated; (iii) a post-project summary including the number of patients reached, patient comments/feedback, and effectiveness of project; (iv) a brief statement for each of the three educational standards addressed by the project to show how the standard was addressed by this HWP; and (v) a 300-word self-reflection using the prompt: ‘In 300 words, reflect upon the importance of the pharmacist to the Health and Wellness of the community, your Lessons Learned, and how this project will influence your future career’. Critical reflection was selected as the method of choice to document student professionalization, as self-reflection is an essential skill in self-directed lifelong learning and enables students to develop confidence in their abilities, learn from their experiences, and discuss any transformative changes they wish to make in their future practice [16,17,18].

### 2.2. Theoretical Framework: Self-Determination Theory (SDT) 

Ten Cate et al. challenges those in health professions education to utilize the framework of the self-determination theory [SDT] when considering learning experiences for trainees [19]. In the 1980s, SDT was introduced by psychologists Ryan and Deci [20] as a theory to describe one’s engagement and motivation in an activity, including (but not limited to) health and wellness behavior change or learning. The tenets of the theory underpin how an individual (the patient or learner) moves along a spectrum from amotivation to extrinsic motivation to intrinsic motivation, the latter representing the most autonomous state [19,20]. To support individuals along this spectrum, three fundamental needs must be met: (i) competence, the ability to demonstrate mastery and have an effect or influence on surroundings; (ii) relatedness, having connections with and care for others; and (iii) autonomy, the ability to self-organize experiences and activities. In both health promotion and learning, the more fully intrinsically and autonomously motivated the individual becomes, the greater the achievement of health or learning outcomes [19]. In the literature, the SDT framework is also applied to the professional identity formation (PIF) of pharmacy students, an evolutionary process in which student pharmacists begin to think, act, and feel like pharmacists. Progressing from extrinsic to intrinsic motivation, coupled with feelings of competence, relatedness, and autonomy in one’s professional role, strongly supports student PIF [21,22]. Thus, it is advantageous to view health professional student learning opportunities through the lens of SDT to optimize the professional development value of the experiences [19,23]. 

### 2.3. Data Collection and Analysis 

Datasets from the 2022–2023 study cohort were de-identified for analysis. A multi-methods design of quantitative and qualitative assessment techniques to analyze the impact of creating/implementing the HWP on students’ learning and professional development was applied. This included (i) quantitative numerical results of instructor-graded rubrics, (ii) a description of the types of projects conducted and standards addressed, and (iii) a qualitative thematic analysis of student reflection/narratives. For the quantitative analysis, an EE course instructor for the Community APPE evaluated each HWP submission using a rubric (which examined project description, materials created, standards addressed, post-project summary, and reflection narrative) and assigned an overall rating of exemplary or acceptable. While ‘unacceptable’ was a possible rating on the rubric, the course instructor would return any summary deemed unacceptable to the student and request a resubmission until the project summary was deemed acceptable, and thus, all project summaries herein are evaluated as either exemplary or acceptable. To view the complete rubric, see Appendix A.

To gain a more nuanced perspective on this learning experience, students’ assignment responses and self-reflection narratives were thematically analyzed. For the qualitative analysis, six submissions per each of the eight APPE modules in the study period (48 total) were randomly selected using the RAND function in Excel. Randomization of projects allowed for a more objective and authentic analysis, limiting possible selection bias. Selected projects were de-identified and loaded by author CT into an Excel database for analysis, which occurred in two phases [24]. In phase one, two investigators (JC, MV) separately reviewed all 48 submissions using memoing to create an initial list of potential codes for what students learned (value of the project on student learning) and would carry into or apply to their careers (impact on professional development). These two investigators then used this preliminary list of codes to assess 5 essays and calibrate coding. Discrepancies or challenges encountered were reviewed and discussed with a third author (LLB). Investigators came to a consensus on 24 codes that fell within two broad areas: the value of the project on student learning and the impact of the project on professional development. As the value of the project on student learning reflected the CAPE outcomes [2] that students opted to address within their HWPs as per Table 1, these outcomes were used as the framework for four overarching themes for 11 agreed-upon codes. Students were required to practice and apply the CAPE outcome ‘Promoter’ through the implementation of the HWP; thus, the team agreed that the qualitative coding of this theme only occurred if the student reflections demonstrated a self-determined current or future role in Health and Wellness promotion and the value of that role. 

To assess the HWP impact on student professional development, the 12 remaining initial codes were framed as subthemes using the main SDT themes of competence, relatedness, and autonomy. Using a method from the literature used to assess the learning impact in dietetic students [23], any codes analogous to a tenet of SDT were modified toward that construct. For example, the initial theme of “student increased confidence” was modified to “perceived competence”, and the initial theme of “increased opportunity to build patient relationships” was modified to “increased opportunity to connect with patients”. The team utilized the assistance of an experienced SDT researcher for the application of this framework [25]. In addition to the 24 initial codes, 2 codes were added during the framework creation, as follows: (i) the incorporation of the concept of how the project supported student autonomy and (ii) the incorporation of dissonance, which impacts student professional development [26]. The final coding framework yielded 26 subtheme codes within 9 overarching themes. In phase two, after the establishment of the coding and thematic framework, one investigator (JC) identified relevant text extracts in each reflection and assigned all applicable coding categories to each extract. Crosschecking of 12 (25%) essays by a second author (MV) was completed, followed by team discussion regarding any discrepancies and selection of illustrative quotes to reflect themes.

## 3. Results

This Community APPE HWP exercise was completed by 157 students in the 2022–2023 cohort; 48 students completed a second Community APPE, yielding 205 HWP projects.

### 3.1. Instructor-Graded Rubrics 

All 205 projects met acceptability per rubric evaluation, with 94 (46%) of these projects deemed “exemplary.” Topics of HWPs (number completed) included immunizations (57), medication management (23), diabetes (20), allergies/cold/flu (19), over-the-counter medications (19), infectious diseases (13), cardiology (12), and miscellaneous (42). By default, ACPE Standard 2.3 (Health and Wellness) [13] was covered in every HWP. Table 2 includes the additional standards that were covered in the HWP (each project was to include at least three), accompanied by a description of how the educational outcome was met, as derived from student summaries.

### 3.2. Impact on Student Learning (Center for the Advancement of Pharmacy Education (CAPE Outcomes)

Analysis of reflections indicated student learning in themes directly related to the CAPE outcomes, which was anticipated as students were expected to address the CAPE outcomes in their HWP. Table 3 presents the four overarching themes of impact (with 11 coding subthemes), the raw number of extracts that informed each theme, an explanation of the theme, and illustrative sample quotations as derived from student submissions. 

### 3.3. Self-Determined Professional Role in Health and Wellness (H&W) Promotion

While it was anticipated that students would hone the “Promoter” CAPE outcome skills through the implementation of the HWP project, 65% (*n* = 31) discussed their desire and commitment to take part in H&W promotion as a pharmacist; this eclipsed simply learning this CAPE outcome and demonstrated the HWP influence on professional development. The language used suggested immersion into their community of practice with verbiage such as “we” and “our” when referring to pharmacists, patients, and their community. Students self-identified why they felt motivated to engage in H&W promotion as part of a pharmacist’s responsibility, citing the “value” of this engagement for themselves, their profession, or their patients/community. While value was described in a variety of ways, students cited internal motivations, feeling their efforts made an impact and stating that they enjoyed the activity and found it rewarding to engage:

“Overall, it was rewarding to put the skills and knowledge that I’ve acquired in my coursework to use in practice, and I feel that I am well-prepared to help patients in the community to manage their chronic diseases and improve their Health and Wellness.” (Participant #4) 

Through empowering and supporting patients’ health decisions, students felt their HWP activities improved adherence, prevented disease, and enhanced health care outcomes, providing value to their communities. Thus, students valued their profession as a “powerful tool” in aiding patients toward better outcomes:

“I believe that being able to carry out an H&W project during community rotations allows us, the students, to see the importance of having time to educate patients outside … the dispensing of medications. I believe that it is of vital importance to be able to instruct patients with strategies, recommendations, and Appendix A so that patients obtain better results from their drug therapies and be able to promote lifestyle modifications.” (Participant #30) 

Students also described benefits to “our” profession, which they felt could occur through H&W promotion activities, such as improved relationships and trust with the patients we serve. While most students placed themselves in the community pharmacy setting, some noted how they would carry this important role into practice in other ambulatory settings or into their desired career path in managed care: 

“This project has taught me that there is simplicity in pharmacists promoting the health and wellness of communities. It isn’t difficult for patients to receive the support they are looking for. All it takes is going to their local pharmacist who is always ready to support a patient in need. My future as a pharmacist entails that I value the active role that I have in bettering the health and wellness of the community that I become a part of.” (Participant #13) 

### 3.4. Impact of Health and Wellness Project (HWP) on Student Professional Development Mapped to Self-Determination Theory (SDT) Framework

Table 4 presents the three self-determination theory (SDT) themes of impact (with 13 coding subthemes), the raw number of extracts that informed each theme, an explanation of the theme, and illustrative sample quotations, as derived from student submissions. 

### 3.5. Dissonance 

As dissonance can impact professional development, statements reflecting dissonance in student assignments were reviewed (*n* = 6). One student commented on how they did not learn anymore on their Community APPE other than “backend billing and insurance shenanigans … only relevant if I open up my own pharmacy (Participant #12).” While not related to the assignment, the student noted activities not congruent with their professional identity; however, these important tasks are relevant and essential to the profession where the community pharmacist is rarely compensated for other than a product. Due to the significant impact insurance “shenanigans” have in our profession, here we perhaps missed an opportunity to further student learning. A few students commented about how they appreciate having the time to engage patients, which is not afforded to many community pharmacists in a fast-paced environment (Participant #33) where pharmacists are “too busy” (Participant #36). Two others noted the pharmacist as “underutilized” (Participant #31) and “overlooked” (Participant #43) in their role in the health care system.

## 4. Discussion 

### 4.1. Importance of Health and Wellness (H&W) Promotion in Advanced Pharmacy Practice Experiences/Curriculum

The inclusion of learning activities within the pharmacy curriculum to engage student pharmacists in the promotion of H&W is a crucial element in laying the educational foundation for future practice [1,2,3,4]. Our HWP learning activity placed in the Community APPE provided each student the opportunity to demonstrate the ability to create and implement a health promotions campaign for a target audience. With 100% of students meeting our rubric criteria for an acceptable HWP submission, our students successfully addressed the ACPE Standard 2.3 Health and Wellness outcome [13]. The HWP afforded students an opportunity to gain knowledge, enhance communication skills, interact with patients, and devise strategies to educate a target audience, as shown in Table 3. It is anticipated that many of the skills that students developed, such as developing visual aids to accompany a promotion event or employing empathic listening skills, are transferable such that students now have some professional tools at the ready to create and implement an HWP in future practice. Unique to our learning activity, students created their own projects, as opposed to previous studies in which students were invited to participate in pre-established health promotions [5,6,7,8,9,10,11,12]. By allowing students to create their own projects, students are empowered to apply the creative/innovative mindset (an expectation of the pharmacy curriculum [3]); in addition, allowing students to self-select a topic of interest underpins the development of intrinsic motivation—a critical element on the SDT continuum, at which juncture the individual is likely to be highly self-directed and autonomous (in creating and implementing the HWP) [20].

### 4.2. Impact of Health and Wellness Project (HWP Participation on Student Learning and) Professional Development 

As shown in Table 2 and Table 3, when designing their HWPs, students incorporated several ACPE Standards/CAPE Outcomes in meaningful ways, such as advocacy, health literacy, cultural competency, and patient individualization. This is an important contribution to the literature, as previous studies have indicated a paucity of documentation on specific learning activities within APPEs in reference to advocacy, health literacy, and cultural competency [14,15]. The rich quotations in Table 3 and Table 4 describe in the students’ own words the myriad impacts of the HWP on professionalization. Indeed, the HWP learning activity provided various eye-opening and “aha” moments in which students learned important lessons or strategies to carry forward in practice, including the importance of individualizing patient-centric care and taking the time to educate and empower patients to advocate for their own healthcare. Anecdotally, some students encountered barriers in their project implementation, as mentioned in their project summaries, including limitations in access to and/or knowledge of resources (color printers, posterboard, design software, etc.), low patient volume, lack of “advertising” project in advance, and waiting until the last minute to display the project; students demonstrated learning in their summarization of Lessons Learned on how they would overcome barriers in future iterations of the project.

While all students practiced the role of ‘Promoter’, 65% of them were also self-determined to serve in this future role in their community. Specifically, students self-identified why they felt motivated to engage in H&W promotion as part of a pharmacist’s responsibility, citing the “value” of this engagement for themselves, their profession, or their patients/community. Student immersion into the role of a H&W “promoter” in the future demonstrates an important component of professional development. The self-identification of this role and of the “value” demonstrated the transition from controlled extrinsic motivation (controlled motivation because they were assigned to the HWP) toward internalized extrinsic motivation or intrinsic motivation (such as feeling rewarded on a personal level, feeling enjoyment in the activity) on the SDT spectrum of motivation. This implies the student can see themselves in this role in the future beyond this assignment due to either their own interest or satisfaction in the patient interaction (intrinsic motivation) and/or by truly internalizing the importance or value to the patient, pharmacy, or pharmacist (partially or fully internalized extrinsic motivation). The movement from controlled, external toward internal extrinsic motivation (even without reaching full intrinsic motivation) leads to a higher level of commitment to the behavior and wellbeing [27]. In learning experiences, this movement is associated with higher academic performance [19]. 

Through applying the SDT framework, yet another unique aspect of our study, we provide evidence that the HWP supports students’ learning and professional development by ensuring that the three primary needs of the learner, as described in SDT, are met [19]. Particularly robust statements indicating professional development are shown in Table 4 quotations under competence, relatedness, and autonomy, in which our student pharmacists demonstrate that they are beginning to think, act, and feel like a pharmacist, the underpinnings of professional identity formation [21,22]. Developing a feeling of topic mastery for previously learned or new topics and being able to apply knowledge and skills in a way that felt meaningful or effective while making a direct impact on their community supported students’ feelings of competence. While we did not specifically study the impact of the HWP on its intended audience (as that was not our study’s aim), we believe from reading students’ assignment summaries (see Table 4 quotes) and have anecdotally noted preceptor comments on evaluation documents (not shown or formally assessed): the HWP did have a positive impact on the target audiences who received the educational intervention. Reflections revealed how the outreach project enabled students to create valued relationships with patients and how they felt continuing this outreach in their future careers could help create that relatedness and connectivity among those in their community. Students described connecting to and feeling part of their chosen profession’s community of practice, expressing pride in the trusted role the profession holds within their communities. Having a sense of belonging within a significant community that is valued by others is a component of relatedness [19]. While there were few student reflections alluding to autonomy support within the learning experience itself, the entire project supported the learner’s autonomous role as students were able to self-identify the project, develop, implement, and self-direct their actions with autonomy support from their preceptors and site staff. 

### 4.3. Limitations and Future Directions

There are limitations to our work. As noted in the Results, we found that a few students reflected on dissonance. These comments resonated with the authors due to the longstanding pressures facing community pharmacists, which more recently are making national headlines [28,29]. These students’ comments unwittingly reflected upon the crisis facing community practitioners across the country, including insurance “shenanigans”, lack of time to commit to patient care, and lack of adequate staffing leading to burnout. Students viewed the profession as being underutilized for the knowledge and skillset of the pharmacist and overlooked by the healthcare system. Yet, in contrast, students experienced firsthand that the pharmacy profession was not overlooked by patients. An overwhelming number of student reflections discussed “relatedness” (Table 4)—elaborating on the connection the students felt they (and their profession) had with patients and the positive impact the profession makes in the community. Students noted patients entrusting them to help them at all hours, at no cost to the patient, and with no appointment necessary. The predominant sentiment of the reflections was that students see their pharmacist preceptors and teams delivering on the covenantal pharmacist–patient relationship every day, and they want to be part of that profession. In retrospect, if we had prompted all students to comment on dissonance within the assignment write-up, it is possible more students may have discussed this dichotomy in the profession. It is important that faculty and the community of practice address dissonance, as it is critical to advancing students’ professional identity formation as they transition from student pharmacist to pharmacist [16,30].

An additional limitation of our work was that we did not measure the preceptors’ perceived value of the projects. While many students reflected on the value of the HWP to themselves, the patients, and the profession, as noted above, only one student (Participant #9) reflected upon the impact on the pharmacists’ knowledge and skills: 

“Implementing projects like this for pharmacy interns to carry out while on rotations not only directly benefits the patient population of the pharmacy, but also exposes more pharmacists to the concept of educational intervention to promote health and wellness within the community.” 

In further analyses of the project, it would be worthwhile to incorporate how preceptors felt the project impacted the site in terms of workload, patient impacts, staff impacts, and student level of autonomy. Additionally, we are preparing an updated guidance/tip sheet for preceptors (and students) to provide more information regarding the expectations of the HWP. We are also discussing moving the due date earlier in the module (e.g., around week 4 of 6) to provide more time for students to receive feedback. Further, when reviewing the assignments for how the SDT tenets were supported within the project, it was noted that students used the tenets (perhaps unknowingly) in their efforts at health promotion. Students commented about how they developed their intervention to capture patients’ attention (poster, flyer, brochure) yet give patients the option to choose (autonomy) to participate. Students discussed providing patients with information on the topic to empower and educate them, allowing patients to choose to make health behavior changes without judgment (autonomy). A subsequent review of our curriculum revealed that while pre-pharmacy students learn SDT early in their curriculum in psychology and first professional-year students learn about motivational interviewing (the tenets of which are related to SDT [20,27]), the investigators believe students’ ability to impact H&W promotion would be enhanced by providing students with additional training on SDT and its role in supporting H&W promotion and learning.

## 5. Conclusions

Students’ creation and implementation of a patient-centric Health and Wellness Project during their Community APPE resulted in student learning and professionalization. Qualitative analysis of student reflections revealed themes of impact on learning and included knowledge acquisition, enhancement of communication skills, opportunity for patient-centered interaction, selection of targeted educational strategies, and immersion into the role of health promoter. Students also noted all three components of the self-determination theory (i.e., competence in the field, relatedness to patients and profession, and autonomy in creating the HWP), supporting growth in students’ professional identity formation as students begin to think, act, and feel like pharmacists.

## Figures and Tables

**Table 1 pharmacy-12-00005-t001:** Additional educational standards options to be selected for inclusion in the Health and Wellness Project.

Educational Outcome *	Description of Outcome
Advocacy	The graduate must empower patients to take responsibility for, and control of, their health
Cultural Sensitivity	The graduate must be able to recognize social determinants of health to diminish disparities and inequities in access to quality care
Health Literacy	The graduate must assess a patient’s health literacy and modify communication strategies to meet the patient’s need
Patient Individualization	The graduate will evaluate personal, social, economic, and environmental conditions to maximize health and wellness
Educator/Educational Strategies	The graduate must educate all audiences by determining the most effective and enduring ways to impart information and assess understanding and will select the most effective techniques/strategies to achieve learning objectives
Communication Strategies	The graduate will effectively communicate verbally and nonverbally when interacting with an individual, group, or organization
Technology	Use available technology and other media to assist with communication as appropriate

* = educational outcomes derived from CAPE 2013 [2] and/or ACPE Standards 2016 [13].

**Table 2 pharmacy-12-00005-t002:** Educational standards students self-selected for inclusion in Health and Wellness Projects.

Educational Outcome *	N	Illustrative Students’ Description of How Standard Was Met
Advocacy	81	Students provided patients with a deeper understanding of the importance of medication adherence and disease prevention with the aim of empowering patients to self-advocate for their own health concerns.
Cultural Sensitivity	19	Students interacted with patients from varying cultures and socioeconomic backgrounds and diminished disparities/inequities in access to quality care by providing options for patients to access free services when applicable.
Health Literacy	60	Students’ HWPs utilized image-based, simplified, and/or patient-friendly language to impart relevant information to facilitate patient understanding.
Patient Individualization	35	Students adapted and personalized information for patients through written materials (e.g., for health literacy, visual acuity in seniors, and terminology) and through verbal interactions (e.g., for health literacy, plans to quit smoking, prior knowledge, and patient preferences, such as cost/generic prescriptions, dosage forms, or desire for over-the-counter self-care).
Educator/Educational Strategies	68	Students selected appropriate educational strategies, such as poster presentations to encourage questions from patients, take-home handouts to serve as reminders, large fonts for emphasis in written materials, or visual images accompanied by hand gestures during discussions. Students identified opportunities to improve the HWP intervention, such as workflow, physical space, or process improvements.
Communication Strategies	0 **	Students practiced verbal, non-verbal, and empathic communication skills through patient interactions. Students employed written visual aids (posters/pamphlets) to initiate patient interaction and prompt patients to request more education.
Technology	0 **	Students educated patients on the use of smartphone apps (e.g., for smoking cessation) and how to research different Medicare plans and enroll online. Student pharmacists sifted through myriad sources available on the internet, dispelling myths and providing quality, up-to-date information to patients.

* Students were to select at least 3 standards per project. ** While no student indicated that they included/met this standard in their HWP, their summaries did include references to these two standards and are included as information for the reader. HWP = Health and Wellness Project.

**Table 3 pharmacy-12-00005-t003:** Themes derived from analysis of student reflections on value of Health and Wellness Project on student learning, knowledge, and skill development.

Main Themes (N) (Subthemes)	Explanation	Illustrative Quotations (Participant Number (#)
Knowledge Acquisition (22) (increased knowledge of project topic, topics outside defined project scope)	Students expressed learning in-depth knowledge preparing for their project and appreciated the ability to apply learned didactic knowledge to actual patients. Several commented on selecting a project topic that was either not heavily detailed in the didactic curriculum or was new, facilitating self-directed learning. By increasing patient/community contact, the project exposed students to queries outside their project scope, leading some to appreciate this challenge to their knowledge base. Students learned how to confirm information, research an unknown topic, or look up medication interactions for a specific patient and provide needed follow-up.	“There have already been new diabetes therapies out since I learned about them during my (therapeutics) course...Knowing the current therapies and factoring in individual factors of patients for each of these therapies will help me become a better pharmacist when I start practicing.” (#36) “This project has allowed me to research and learn more information about immunizations than any class during my academic career. I can now use this knowledge as I move forward in my pharmacy career, educating more and more patients.” (#18) “The project I had was very focused on allergies and the medications used to treat them, such as intranasal corticosteroids, antihistamines, and decongestants, but many patients have questions regarding their over-the-counter medications for other areas like pain, digestive health, cough, injuries, dry eyes etc.” (#3)
Communication Skills Enhancement (40) (application of verbal communication skills, adaption of verbal communication to individual patients, role of communication to initiate project, use of technology to enhance communication)	Reflections demonstrated student application of a myriad of verbal and non-verbal communication skills. Students practiced conducting patient interviews to gather past medical/medication histories and discover patient preferences while learning to ask the right questions in effective ways, using a friendly tone with easy-to-understand terminology, and practicing better active listening skills. Students tailored verbal communication to the individual, considering health literacy and prior topic knowledge to “meet the patient where they are.” Several described the verbal and non-verbal (eye contact, smiling) skills needed to draw the patient in for discussion. A few students described the utilization of technology for patient learning, including live animations on slide presentations, portable technology for health/medication topics, or selecting patient-oriented medication adherence aids.	“I feel that this (HWP) will improve how I counsel patients as I can tailor the counseling specifically towards a patient. What may be an important counseling point for one patient may be different for another patient. While it is important to have a mental checklist of counseling points, it is more important to gauge what the specific patient knows.” (#25) “The biggest lesson I learned was patient communication. … I learned how to effectively communicate with different patients. There are some patients more familiar with their health and medical language while others had trouble understanding their medications and health. Once recognizing the extent of the patient’s understanding, I would adjust my language and how I went about explaining and answering their questions.” (#14) “I was able to make eye contact or draw almost all customers’ attention that passed by…. All patients remarked how my poster design and presentation, as well as my persistent eye contact and smile drew them to at least check out what I had to offer at first.” (#43)
Patient-centered care (22) (tailoring recommendations to individual patients)	Students described actively participating in the selection and implementation of recommendations to individual patients, considering person-specific factors such as a preference for self-care, dosage forms, cost considerations, social determinants of health, and cultural factors. Student projects that were not medication or condition-specific but rather focused on either access to medications (selecting Medicare plans, cost savings programs) or adherence also described tailoring or customizing their recommendations to the specific patient needs, determinants, and preferences.	“This experience has allowed me to be more culturally sensitive and learn the demographic of those who are around me. During my rotation, the pharmacy was in an underserved location where many people did not have the resources to pay for the medications. While working on my project, I was able to incorporate/think about the costs of the recommendations. This notion of cultural/demographic sensitivity is something I will carry along with me for the rest of my rotations and pharmacy career.” (#5) “Each patient has different needs and different personalities. For my compliance aide HWP, for patients who were more technology inclined, I could recommend apps on their smartphones and for other patients I would recommend more physical strategies like pill boxes.” (#40) “Patient individualization is crucial when we help each patient. Every single one (who attended my Brown Bag event) had a different economic and social status that affected their medication selection which we have to take into account in order to maximize pharmacological and nonpharmacological therapy.” (#32)
Educator (65) (selecting educational strategy, adapting visual aids (poster, handouts) to the community, the role of visual aids to initiate interaction, reflecting on opportunities for improvement)	Preparing the project, students considered a wide range of factors, such as choosing or creating the visual aids to be used (flow charts, infographics, posters, handouts) and how to adapt them to their desired targeted audience (literacy, age, visual acuity). They considered which visuals would “catch the interest” of the shopper or patient to garner participation. Many reflected on ways they could expand or enhance the project either through the materials provided, its placement in workflow, the physical location of the intervention, or the selection of the targeted audience demonstrating learning.	“The project met the educator standard by using a plethora of visuals as summaries for patients. A table comparing symptoms of the flu and cold was used as a quick summary to differentiate the two conditions. Large visuals were used to attract patient attention such as an image of a Tylenol bottle which was easily recognizable… Patients commented that the visuals attracted their attention such as the large Tylenol image which gave patients a sense of familiarity… I learned that a simple poster board can be enough of a reminder to get patients to act.”(#25) “I determined the most effective and enduing ways to impart information was to display my poster in the waiting area of the pharmacy. Most patients have to wait for their prescriptions to be ready and while they would wait I’d notice they would read my poster and have follow-up questions. … If I noticed a patient reading my poster I’d ask if they have any questions or interested in receiving a vaccine.” (#20) “I think if I were to do this project again I would change the theme of the table every three days or so. This would offer the community more information on other topics that they may have been more interested in. Mental health has a stigma around it, so some saw that sign and were immediately uninterested.” (#6)

**Table 4 pharmacy-12-00005-t004:** Themes derived from analysis of student reflections on impact on professional development using self-determination theory framework.

Main Themes (N) (Subthemes)	Explanation	Illustrative Quotations (Participant Number (#)
Competence (66) (perceived increased competence in knowledge/skills, received positive reinforcement from patients, positive preceptor/team feedback, perceived direct impact on patient care, identified need for continued professional competence)	Students described increased confidence in their knowledge and skills, deriving motivation from the positive feedback received from both patients and their pharmacy teams. Many felt their projects made an impact in their community, and often, they were able to immediately identify the impact of their interventions when patients received recommended vaccines (or made appointments), selected an OTC therapy, made behavioral changes, or reported increased knowledge on the topic. Students identified a personal need for and commitment to lifelong learning, identifying the responsibility pharmacists have to their patients to provide up-to-date, accurate information.	“This project will positively influence my future career because it instilled the confidence in me and my education.” (#17) “By conducting this project, I made a direct impact on improving patients’ overall wellness by encouraging tobacco cessation and assisting patients in developing quit plans. I was able to confidently walk patients through the various medications available, compare the pros and cons of each, and assess which best fit the individual’s preferences. …. This project will influence my future career because it has helped me to feel confident in my abilities as a patient educator. Additionally, I learned specific strategies that I can employ to be a more effective educator.” (#4) “This project will influence my future career as a pharmacist because it enforces the idea that being a pharmacist means that you will never stop learning. … we have to keep ourselves up to date. We are often the first-place patients will go to ask questions because we are so easily accessible. This has encouraged me to always be aware of changes to guidelines and medical news in general so I can help my patients the best I possibly can.” (#31) “This project has influenced my future career because I am ready and confident to join community pharmacists … [it] has been a wonderful capstone and segue to my future career.” (#43)
Relatedness (59) (increased opportunity to connect with patients, felt connected to profession, felt entrustment from patients, appreciated pharmacist valued role in the health care system)	The HWP increased opportunities to establish relationships with, and care for, patients in the community. Students related to their community of practice by identifying role models and connecting with pharmacy team members. Many reflected upon and accepted responsibility for the high level of trust the community places upon their pharmacist, with a few students being “shocked” and awed by this level of trust. Students demonstrated a strong appreciation for being a part of the significant role the pharmacist holds within the health care system, often the patient’s first stop in their health care journey. The value patients place on the pharmacist is not solely due to the ease of access (no cost, no appointment, various hours of the day) but because patients rely on the pharmacist as a trusted source of accurate information who weeds out false information, triages patient needs and bridges a knowledge gap left by the internet or other health care providers.	“This project allowed me to really work with individuals to help them with next steps and potential options to provide them some sense of hope, and perhaps most importantly let them know that someone does care and is willing to listen to and validate them.” (#46) “What truly makes a pharmacist so important to the health and wellness of the community is their vast knowledge of medicine as well as their accessibility to the community. Activities like this make me love more and more my career as a pharmacist, we can be extremely helpful with little things that can be very simple for us and mean so much to patients and even prevent them from getting exposed to possible harm.” (#32) “All of the patients I was able to talk with and counsel were so grateful for the help I was able to provide. This project helped reinforce the foundation of why I chose to attend pharmacy school 6 years ago, which is the drive to help people.” (#21) “During this rotation, my preceptor had remarked that over the counter medications essentially provide pharmacists with the opportunity to prescribe on a smaller setting. This responsibility to guide patients toward products should not be taken lightly as patients trust pharmacists to recommend the best product for them and their specific scenario… I feel that this project helped instill in me how much trust and appreciation individuals have toward their pharmacist.” (#21)
Autonomy (24) (pharmacist promotion of patient self-advocacy, student use of autonomy support concepts, the impact of the project on student autonomy needs)	Students describe either their own or their pharmacist’s role in promoting patient self-advocacy, seeing the value of autonomy in making health behavior change. Assignments reflected the use of autonomy-supporting principles, such as providing patients with accurate information to make their own health choices, asking patients for their goals, avoiding judgment in verbal or non-verbal communication, and avoiding pressure tactics. Several students crafted their projects with the mindset of allowing patients to choose to participate by asking questions about the HWP materials presented in the visual prompts of a poster or pamphlet.	“...Patients were empowered to create their own quit plan, rather than being given a “one size fits all” quit plan. I assisted patients in creating their own quit plan. … One patient noted that it was more encouraging and empowering to think about the benefits associated with making a change rather than the consequences associated with continuing smoking.” (#4) “When talking to people about my project, especially those who aren’t ready to quit, I would communicate the information in a way that didn’t make them feel pressured and reenforce that it is completely their decision.” (#14) “In my future career I will always take the time to talk, communicate, and educate any patients wanting to know more information on their vaccine status. I will be supportive in the decisions they make and serve as an aid to beneficial decision making.” (#47) “With this project I was able to provide patients with the education to make appropriate choices in their own treatment and use medications appropriately. By setting up a poster for patients to read, I gave them the opportunity to learn about over-the-counter medications without making them go out of their way or inconvenience themselves to learn. I empowered them to take control of their own learning without forcing it upon them.” (#45)

HWP = Health and Wellness Project; OTC = over-the-counter.

## Data Availability

Data are contained within the article.

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
