# Peer review of "Health and Wellness Projects Created by Student Pharmacists during Advanced Pharmacy Practice Experiences: Exploring the Impact on Professional Development"

_pharmacy, 2023, doi:10.3390/pharmacy12010005_

Round 1

Reviewer 1 Report

Comments and Suggestions for Authors

Thank you for the opportunity to review this manuscript. This is very well written and relevant article for the journal. It provides detailed information.

Abstract

More information needs to be added about the context needs to be added to the background i.e the program PharmD in the USA, 4th professional year.

It appears this is a multi-method rather than a mixed methods study. As there is no integration and also the specific mixed methods study design is not mentioned (i.e. convergent, explanatory, exploratory etc…) if that is the case then please update abstract to reflect.

Introduction

Suggest moving 1.2 Self-determination Theory (SDT) under methods with a subheading theoretical framework. If moved them the aim should be in the last paragraph of the introduction.

Methods

As above this appears to be a multi-methods rather than a mixed methods study as the design was not described and there is no mention of integration in the data analysis.

2.2 Data collection and analysis. How were the students reflections “randomly selected”(i.e. every reflection had the same opportunity to be chosen). Whey not select this purposively? What was the rationale? Can the implications of this be discussed.

Minor line 121 add US i.e College of Pharmacy in the United States.

Minor line 98:  CAPE please spell the acronym the first time you use it (line 98) Center for advancement of Pharmacy Education (CAPE)

Results

Do the students have access to previous projects? Are there areas/suggestions provided to them? Is the something in the reflection that gives information about motivation behind the project?

Is there tension between the rhetoric and the reality. I.e what the students thought they had achieved vs what really happened in practice?

I was surprised that the students didn’t describe any barriers difficulties in their reflection. Does this mean that all projects were successful (i.e. all had patients’ interested/involvement)

Discussion

As queried above. Did the students reflect on the difficulties on funding and implementing their projects and how they overcame these?

What are the eye-opening” aha” moments the authors refer to. Can a couple of examples be provided to illustrate this point.

Can you discuss the so-what of the dissonant results. I found this extremely interesting.

Suggest adding a conclusion.

Reviewer 2 Report

Comments and Suggestions for Authors

REPORT

Health & Wellness Projects Created by Student Pharmacists during Advanced Pharmacy Practice Experiences: Exploring the Impact on Professional Development

BRIEF SUMMARY:

This study investigated the effect on student learning and professional development when Pharmacy students created, implemented, and reflected on a Community APPE Health and Wellness Project of their choice.

GENERAL COMMENTS:

·         Writing and use of language is excellent making the manuscript generally easy to read.

·         The abstract is succinct, clearly explains the work conducted for quick reference if needed.

·         The use of clearly presented tables in the results section enables the reader to easily assimilate a considerable amount of information.

·         However, in Method, the section ‘2.2 Data Collection and Analysis’ (line 152 – line 179) is complex and, at times, quite difficult to follow.  Could this also be presented in table format to show flow of steps followed?

·         A very interesting and important study.  I read the paper several times to fully understand the wealth of information it contained on innovations in student learning and professional development.

·         A great deal of work has been put into this paper to ensure accuracy of information and very interesting presentation. I have no additional comments to make.

SPECIFIC COMMENTS:

·         Line 169: change ‘dietitian’ to ‘dietetic’ students.

Reviewer 3 Report

Comments and Suggestions for Authors

Thank you for the opportunity to review this manuscript. Overall, I recommend it for publication as it provides information on the operations and benefits of a student-led health and wellness project. My main concern centers on the decision to report only on the reduced codes. The methods identify that an inductive coding process was used, however the results read more like a deductive coding process. This is not inherently a major concern. As qualitative research, the reader might benefit from understanding the original 24 codes that were "induced" from the thematic analysis. Consideration could be given to a table showing the original 24 codes with sub-categorization to the segments used to provide the results and analysis. This may aid the readers in determining how they might implement projects in their own colleges, or modify accordingly. This isn't a required modification for publication, but one that may upgrade the value.

Reviewer 4 Report

Comments and Suggestions for Authors

Thank you for organizing such a project for students. some suggestions:

I would suggest to include references from FIP (Global competency Framework), especially on Public health and Pharmaceutical care services.

I would also suggest to classiffy (group) ideas for implementation at the end of the manuscript.

It would be good to explore possibilities of implementation in real-life setting and compare conclusions with current practices.
